# Active tactile discrimination is coupled with and modulated by the cardiac cycle

Alejandro Galvez-Pol[1,2]*, Pavandeep Virdee[1]*, Javier Villacampa[2], James Kilner[1]*

[1]Department of Clinical and Movement Neurosciences, Queen Square Institute of Neurology, University College London, London, United Kingdom; [2]Psychology Department, University of the Balearic Islands, Palma de Mallorca, Spain

**Abstract** Perception and cognition are modulated by the phase of the cardiac signal in which the stimuli are presented. This has been shown by locking the presentation of stimuli to distinct cardiac phases. However, in everyday life sensory information is not presented in this passive and phase-locked manner, instead we actively move and control our sensors to perceive the world. Whether active sensing is coupled and modulated with the cardiac cycle remains largely unknown. Here, we recorded the electrocardiograms of human participants while they actively performed a tactile grating orientation task. We show that the duration of subjects' touch varied as a function of the cardiac phase in which they initiated it. Touches initiated in the systole phase were held for longer periods of time than touches initiated in the diastole phase. This effect was most pronounced when elongating the duration of the touches to sense the most difficult gratings. Conversely, while touches in the control condition were coupled to the cardiac cycle, their length did not vary as a function of the phase in which these were initiated. Our results reveal that we actively spend more time sensing during systole periods, the cardiac phase associated with lower perceptual sensitivity (vs. diastole). In line with interoceptive inference accounts, these results indicate that we actively adjust the acquisition of sense data to our internal bodily cycles.

## Editor's evaluation

This important study investigates the relationship between touch perception and the cardiac cycle. The authors show that people spend more time touching a surface during cardiac systole, where tactile perceptual sensitivity is relatively poor. These findings provide convincing evidence that people actively adjust how they sample sensory information based on internal body states.

*For correspondence:
a.galvez-pol@uib.es (AG-P);
pavandeep.virdee.16@ucl.ac.
uk (PV);
j.kilner@ucl.ac.uk (JK)

**Competing interest:** The authors declare that no competing interests exist.

## Introduction

Sensations often arise through the active movement of the sensor apparatus (e.g., fingers and eyes) rather than through the passive movement of the stimulus. This notion was already noted by *Gibson, 1962*, who in the context of active sensing proposed that active touch refers to what is ordinarily called touching (vs. passive touch, or being touched). Also, he proposed that active touch is built upon the processing of our own body in space and the stimuli in our environment, that is, proprioception and exteroception. Yet, it is now well established that sensations also depend on the internal state of the body when the stimuli are processed (i.e., interoception; *Al et al., 2020*; *Azzalini et al., 2019*; *Galvez-Pol et al., 2020a*; *Garfinkel et al., 2014*; *Khalsa et al., 2018*; *Quigley et al., 2021*; *Salomon et al., 2016*). Our study aimed to test whether active touch sensing varies a function of phasic changes inside the body.

The most studied source of interoceptive signalling is the heart. As an intrinsic oscillator, the heart has two phases: in the *systole* phase the heart ejects the blood, whereas in the *diastole* phase it refills.

**eLife digest** Most of what is known about human senses comes from experiments under laboratory conditions where participants stay still and stimuli are presented to them by the scientists. However, this approach does not reflect what happens in real life as we move around, changing the position of our eyes, heads and hands, to actively sense the world.

Our perception also changes depending on what is going on inside our bodies and minds at any one time. For instance, our sensitivity to touch varies during the two phases of our heartbeat: people are less perceptive to being touched during systole (when the heart ejects blood), compared to when they are touched during diastole (when the heart refills with blood). But it was unclear if this relationship influences how we actively touch and sense objects. For instance, do people seek touch in a particular phase of their heartbeat, and how does this change their response to the object?

To investigate, Galvez-Pol et al. traced people's heartbeats while they actively touched different objects. Without looking, the participants had to work out whether the objects had vertical or horizontal grooves. Although they did not start their touches in a specific phase of the heartbeat, their hearts did influence their behaviour. If they started the touch during systole, they held their fingers over the object for longer. The effect was especially noticeable when it was difficult to discriminate the objects' grooves.

Galvez-Pol et al. reasoned that this was down to participants having to compensate for the loss in touch sensitivity during the systole phase of their heartbeat. This suggests that people actively adjust how they acquire sensory information, such as touch, based on how their bodily functions alter their senses.

These findings provide a starting point for future studies investigating how internal bodily fluctuations impact how we sense and respond to things in real world scenarios. This could potentially shed light on the differences between the way neurotypical and neurodivergent individuals sense the world.

Both phases together comprise a full cardiac cycle. Previous studies have shown that participants' responses to stimuli vary according to the phase of the cardiac cycle in which exteroceptive stimuli are presented. Perceptual sensitivity to painful, visual, tactile, and auditory stimuli are typically reduced when they are presented in systole compared with diastole (*Al et al., 2020*; *Edwards et al., 2001*; *Edwards et al., 2007*; *Grund et al., 2022*; *McIntyre et al., 2006*; *Motyka et al., 2019*; *Pramme et al., 2016*). It has been proposed that these effects reflect cardiac-related sensory attenuation due to the competing allocation of attentional resources (*Berntson and Khalsa, 2021*; *Critchley and Garfinkel, 2018*; *Khalsa et al., 2018*) during periods of afferent signalling from the baroreceptors in systole (*Critchley and Harrison, 2013*; *Garfinkel et al., 2014*). Overall, the current framework proposes that perceptual sensitivity is reduced during the systole phase of the cardiac cycle due to concurrent and transient cardiac-related afferent signals, whereas it is heightened during the diastole phase of the cycle.

The effects of the cardiac cycle and its phases on the processing of exteroceptive information have been shown by presenting brief stimuli to participants. These stimuli are timed to occur during the participants' systole or diastole phases. However, in our everyday lives, exteroceptive stimuli are highly unlikely to be presented to us passively in a cardiac phase-dependent manner. Here and in our previous work, we have adopted a more ecological approach in which participants actively seek information through the movement and control of the sensor apparatus (freely moving to seek sense data, i.e., active sensing). In this context, we have recently shown that in an active sampling visual paradigm, saccades, and visual fixations are coupled to the systole and diastole phases of the cardiac cycle (*Galvez-Pol et al., 2020b*). These results are consistent with the hypothesis that interoceptive and exteroceptive processing adjust to each other by sampling the environment during the most quiescent periods of the cardiac cycle (visual fixations during diastole). In the present study, we expand upon this and other work in active vision (*Kunzendorf et al., 2019*; *Ohl et al., 2016*) to test the prediction that when we move to touch objects, we do so in a manner that reflects the relative perceptual sensitives of touching in systole and diastole.

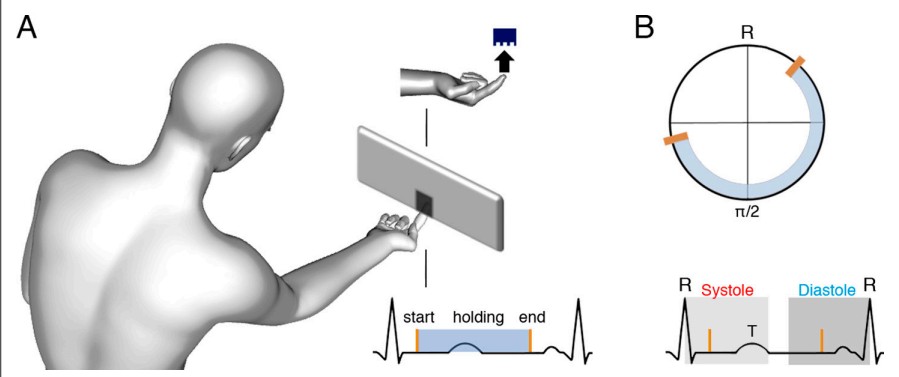

**Figure 1.** Task and schematic illustration of one trial with cardiac and tactile events. (**A**) The participants' task was to touch gratings (one per trial) with the index finger to determine their orientation (vertical or horizontal, probability of each orientation = 0.5). There were seven levels of difficulty according to the gratings' widths. Also, there was a movement control condition with a flat stimulus and no orientation judgement required. Participants were free to start, hold, and end the touch when they felt like, and their electrocardiograms (ECGs) were co-registered. (**B**) Upper panel: for each touch, we computed the start, stationary hold, and end of touch in degrees relative to the entailing heartbeat: for example, for an R–R interval of time $t_R$, where the touch started at time $t_E$, we calculated $t_E/t_R \times 360$. Then, each type of event was subjected to circular averaging. Lower panel: we also computed the proportion and duration of touches starting and ending in the systole and diastole phases of the cardiac cycle. We used the systole length of each cardiac cycle to define a time window of equal length (to equate events' probability in diastole; see e.g., **Al et al., 2020**), which we located at the end of the cardiac cycle.

While active touch sensing has been studied in animals and humans (**Grant et al., 2014**; **Olczak et al., 2018**; **Prescott et al., 2011**; **Schaefer et al., 2009**; **Vega-Bermudez et al., 1991**), no accounts of interoceptive signalling originating from the heart have been considered. Here, we tested the hypothesis that the timing of active touch sensing would be coupled to the distinct phases of the cardiac cycle, and that such coupling would be linked to changes in participants' responses. To test this, we instructed our participants to perform an active tactile discrimination task of grating orientation. We co-registered the participants' electrocardiograms (ECGs), the initiation, stationary holds (to sense the stimuli), and the end of participants' touches. Then, we computed the period and phase of the cardiac cycle in which each touch occurred, as well as the behavioural consequences of sensing the stimuli in the different phases of the cycle.

As previous studies in passive sensing have shown that tactile sensitivity is reduced during the systole phase (**Al et al., 2021a**; **Grund et al., 2022**; **Motyka et al., 2019**), we examined whether this cardiac-related modulation affects the timing and duration of active touch. Specifically, we tested two hypotheses. First, whether the time of onset and/or offset of the touches reflects sampling of the stimulus during periods with greater tactile sensitivity. In other words, do subjects preferentially touch during the diastole phase of the cardiac cycle? Second, we tested whether the duration of active touch is modulated by the phase of the cardiac cycle in which the touch is initiated. The prediction was that because of the reduced tactile sensitivity in systole, touches initiated in this phase would be longer than those initiated in diastole.

## Results
### Overview

To test these hypotheses, 46 participants performed the active tactile discrimination task illustrated in *Figure 1*. Participants' task was to touch a grating with their index finger (one per trial) and determine whether the grating was in a vertical or horizontal orientation. The probability of any given orientation was 0.5. There were seven levels of difficulty that varied as a function of the gratings' widths. In addition, the participants performed a control movement condition with a flat stimulus, to study the simple effects of movement. In this condition, they followed the same instructions, but with no need to report any orientation at the end of the trials. Participants were free to start, hold, and end the touch when they felt like. Throughout the task we recorded the participants' ECGs.

To examine the statistical relationship between when participants touched and the cardiac cycle, we analysed the data in two different ways. First, circular statistics were employed to exploit the iterative nature of the cardiac cycle. We calculated the start, the mean holding phase, and the end of each touch as a function of the whole cardiac cycle. We tested separately whether the participants' mean phase (as a function of the whole cycle) differed from a uniform distribution using Rayleigh tests (e.g., *Al et al., 2020*; *Galvez-Pol et al., 2020a*; *Kunzendorf et al., 2019*; *Motyka et al., 2019*; *Ohl et al., 2016*). However, this analysis does not consider the biphasic nature of the cardiac cycle (systole and diastole phases). Yet, previous studies have shown that responses to stimuli vary as a function of the phase of the cardiac cycle in which information is processed (see e.g., *Garfinkel et al., 2014*; *Al et al., 2020*; *Leganes-Fonteneau et al., 2020*; *Grund et al., 2022*). Therefore, we also examined participants' responses as a function of the phase of the cardiac cycle. We analysed the proportion of events (start and end of touches) in two time windows that encompassed the systole and diastole phases of the cardiac cycle (see e.g., *Al et al., 2020*; *Grund et al., 2022*; *Motyka et al., 2019*), as well as the duration of touches as a function of the cardiac phase in which they were initiated (see Materials and methods for further details).

## Descriptive statistics

The proportion of rejected trials (see Methods) comprised 5.5% of the total number of trials, standard deviation (SD) = 4% across participants. The mean number of computed heartbeats during participants' touch was 540, SD = ±173. Participants' mean heart rate was 76 bpm, with a mean time between R-peaks of 793 ms and SD = ±110 ms. The percentage of correct responses was 82.7%, SD ± 6.6%, and the mean duration of participants' touch in the whole experiment was 1103 ms, SD = 457.40 ms, Mdn = 999 ms. The mean duration of the systole phase was 342 ms and the SD = 23.4 ms, which is the expected duration for the mean heart rate of our sample (*Mao et al., 2003*).

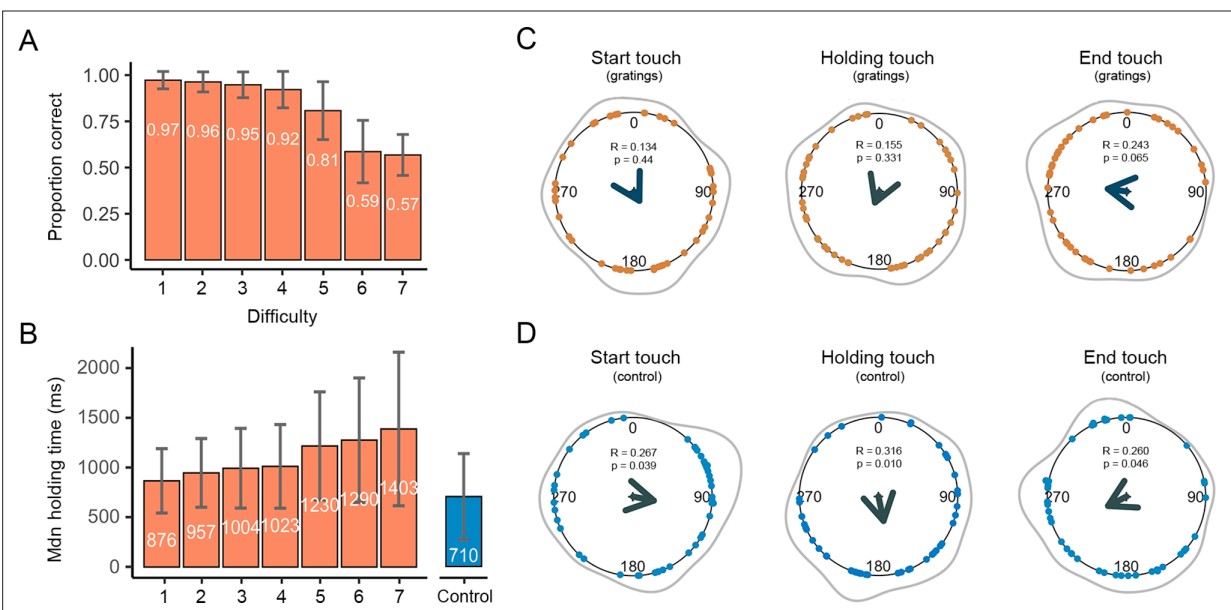

**Figure 2.** Duration, correct responses, and touches as a function of the whole cardiac cycle. (**A**) The mean proportion of correct responses changed as a function of task difficulty. Performance in the first four levels of difficulty was similar and significantly greater than in the remaining levels of difficulty (see main text). (**B**) The duration of participants' stationary holds for the first four levels of difficulty was similar and significantly shorter than in the remaining levels of difficulty. The duration of stationary holds in the control movement condition (flat stimulus) was significantly shorter than those of the gratings condition (see main text); error bars show standard deviations (SDs). (**C**) Circular means showing the distribution of the start, holding, and end of touches for the grating stimuli across the cardiac cycle (interval between two R-peaks at 0/360°). Points depict subjects' mean degrees, the central arrows in v-shape point to the overall mean degree and its length indicates the coherence of individual means. The grey outer lines depict the circular density of individual means. Overall, the start, mean point of stationary hold, and end of touches occurred at 151°, 202°, and 278°; n = 46 (**D**) Touch data for the control movement condition as analysed and depicted in panel C. Here, the start, mean point of stationary hold, and end of touches occurred at 101°, 168°, and 243°; n = 45.

## Proportion of correct responses and touch duration

In an initial analysis, we examined whether the participants' performance was modulated by task difficulty in the gratings condition. To do this, we compared the proportion of correct responses for each level of difficulty (seven levels) with a repeated measures of analysis of variance (rmANOVA). This analysis showed a significant effect of task difficulty on participants' performance ($F$(3.34, 150) = 154, p < 0.0001, $\eta p^2$ = 0.774). Post hoc comparisons showed that participants' performance for the first four levels of difficulty was fairly similar (all p > 0.086; see *Supplementary file 1A, B* for all post hoc comparisons). Yet, performance in these levels was significantly greater than in the remaining levels of difficulty (all p < 0.0001, $d$ > 0.833). Likewise, the proportion of correct responses in difficulty level 5 was significantly greater than those in difficulty levels 6 and 7 (both p < 0.0001, $d$ > 1.618; *Figure 2A*).

Next, we examined the length of time that participants spent holding their touch for each level of difficulty, that is, time spent sensing gratings with different widths. Here, we took the median of participants' holding touch in milliseconds. There was a significant effect of task difficulty on the time spent holding the touch ($F$(1.70, 76.6) = 25.1, p < 0.0001, $\eta p^2$ = 0.358). Post hoc comparisons showed that the length of holding time for the first four levels of difficulty was similar (all p > 0.065), as well as shorter than in the remaining levels of difficulty (all p < 0.01, $d$ > −0.416). Lastly, we compared the length of holding time touching all gratings and the flat stimulus, where no discrimination was required. The results show that the holding time for the control movement condition (flat stimulus) was significantly shorter than for the gratings across all seven levels of difficulty (p ≤ 0.05, $d$ > −0.416; *Figure 2B*).

## Start, holding, and end of touches as a function of the whole cardiac cycle

Studies in active sensing have shown the start of movements to acquire sense data occurs during the early period of the cardiac cycle, whereas the actual sensing has been observed during the mid and later periods (*Kunzendorf et al., 2019*; *Galvez-Pol et al., 2020b*). In addition, studies in passive sensing have shown greater perceptual sensitivity during these latter periods of the cycle (*Al et al., 2021b*; *Grund et al., 2022*; *Motyka et al., 2019*). Hence, we predicted that the start of active touches would occur in the early period of the cardiac cycle, and the sensing touch (i.e., stationary hold) would preferentially occur during the later and more sensitive period of the cardiac cycle. To examine this, we analysed the coupling of the participants' responses (start, mean contact point of the holding touch, and end of touch) with their own heartbeats. We computed the circular phase of these tactile events as a function of the whole cardiac cycle. Then, we used circular averaging for all trials, as well as separately for each level of difficulty.

First, for the gratings condition we pooled the data from touches across all difficulty levels. The start of touches occurred on average during the first half of the cardiac cycle at 151°. The phase of the initiation of touches in circular space did not differ significantly from a uniform distribution (Rayleigh test $z$ = 0.134, p = 0.44; *Figure 2C*). The mean phase of the holding touches occurred on average circa the mid part of the cycle at 202°. Yet, it should be noted that holding touches often spanned across more than one cycle. Their distribution did not differ from uniformity (Rayleigh test $z$ = 0.155, p = 0.331). Likewise, the distribution of ending touches in circular space, which occurred on average during the latter part of the cycle at 278°, did not differ significantly from a uniform distribution (Rayleigh test $z$ = 0.243, p = 0.065). Second, we analysed all trials separately for each level of difficulty. None of the analyses reached significance after correcting for multiple comparisons using the Holm–Bonferroni method for the seven comparisons.

Next, we examined the same tactile events when the participants were instructed to touch the flat stimulus. Thus, no orientation judgement was required, only movement. In the control movement condition, the start of the touches occurred on average during the first half of the heartbeats at 101°. The distribution of these touches differed significantly from a uniform distribution (Rayleigh test $z$ = 0.267, p = 0.039; *Figure 2D*). Regarding the holding of touches, they occurred on average during the mid-part of the cycle at 168° and their distribution differed significantly from a uniform distribution (Rayleigh test $z$ = 0.316, p = 0.010). Last, the end of touches occurred on average during the second half of the heartbeats at 243°. The distribution was marginally different from a uniform distribution (Rayleigh test $z$ = 0.260, p = 0.046).

These results indicate that active touches were coupled with the cardiac cycle in the control movement condition. Similar findings have been found in active visual tasks that did not require perceptual discrimination (see *Kunzendorf et al., 2019*; *Galvez-Pol et al., 2020a*). In a separate analysis of the gratings condition, we did not find evidence of coupling when our participants sought to sense and discriminate the gratings. Yet, while the present analysis considers the iterative nature of the cardiac cycle, it does not reflect its true biphasic physiological nature (the presence of two cardiac phases: systole and diastole) nor differences in how participants performed the task. We examined these in the following analyses.

## Proportion of touches starting and ending in each cardiac phase

To study the coupling between the movement of the sensor to acquire sense data and the cardiac phase, we examined whether our participants started and ended their active touches in a specific cardiac phase. Based on previous research, we hypothesized a greater proportion of touches starting in systole, and a greater proportion of touches ending at the end of diastole; the latter being the cardiac phase with greater perceptual sensitivity (*Galvez-Pol et al., 2020b*; *Kunzendorf et al., 2019*; *Ohl et al., 2016*; *Al et al., 2020*; *Motyka et al., 2019*). Importantly, since the proportion of touches in each cardiac phase reflects and depends on the proportion in the other phase (see Data analysis), we only analysed the proportion of touches in one phase (in systole). Here, two different set of analyses were performed.

First, we compared the proportion of touches that started and ended in each cardiac phase between our experimental conditions. Since the control movement condition had no difficulty level (no orientation to-be-discriminated), we compared it against the gratings data collapsed across all levels of

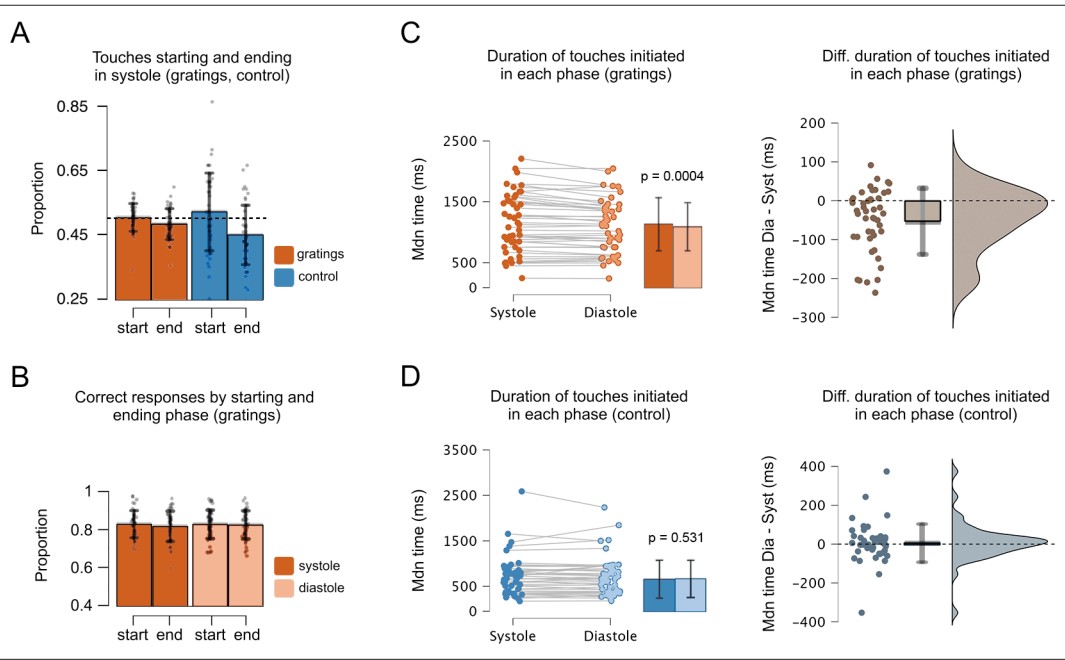

**Figure 3.** Proportion of touches, correct responses, and duration as a function of the cardiac phase. (**A**) In the grating and control movement conditions, the proportion of touches starting in systole was greater than that of touches ending in systole. Also, the proportion of touches ending in systole was significantly smaller than the chance level (0.5, depicted with a dashed line); see main text. Hence, more touches ended in diastole. (**B**) In the gratings condition, the proportion of correct responses was similar regardless the touches were started or ended in systole or diastole. (**C**) In the grating condition, the duration of touches was greater when participants started to touch in systole vs. diastole ($M_{diastole}$ = 1143, $M_{systole}$ = 1093). (**D**) Conversely, for the control condition, the duration of touches was similar when participants started to touch during systole and diastole ($M_{diastole}$ = 708, $M_{systole}$ = 717). Right panels show the difference between the duration of touches (stationary holds) initiated in diastole and systole, that is, for each participant, the holding time of touches started in diastole minus touches started in systole. Error bars show SDs; n = 46 gratings condition, n = 45 control movement condition; *p*-values adjusted using the Holm–Bonferroni method.

difficulty. Here, we performed an rmANOVA with two variables and two levels: experimental condition (gratings, control) and event type (starting and ending touches in systole). This analysis showed a significant main effect of event type ($F$(1, 44) = 9.91, p = 0.003, $\eta p^2$ = 0.184), a non-significant effect of experimental condition ($F$(1, 44) = 0.47, p = 0.499, $\eta p^2$ = 0.010), and a non-significant interaction between both variables ($F$(1, 44) = 4.04, p = 0.051, $\eta p^2$ = 0.084). However, we found a stronger trend for the control movement condition (see *Figure 3A*). For the gratings, the proportion of touches that started in systole was significantly greater than the proportion of touches that ended in systole ($t$(45) = −2.22, p = 0.032, $d$ = −0.327, 95% confidence interval [CI] mean difference [−0.040, −0.002]). Likewise, in the control movement condition, the proportion of touches that started in systole was significantly greater than that of touches that ended in systole ($t$(44) = −2.75, p = 0.009, $d$ = −0.410, 95% CI mean difference [−0.125, −0.019]).

The previous analysis does not test whether the proportion of touches in each cardiac phase is different to what would be expected by chance. We examined this by comparing the proportions of touches that started and ended in each phase against chance (0.5) using one-sample *T*-tests. For the gratings condition, the proportion of touches starting in systole did not differ from chance level ($F$(1, 45) = 0.19, p = 0.662, $\eta p^2$ = 0.004). In contrast, the proportion of touches ending in systole was significantly smaller than chance level ($F$(1, 45) = 6.33, p = 0.015, $\eta p^2$ = 0.123). Similarly, in the control movement condition the proportion of touches starting in systole did not differ from chance level ($t$(44) = 1.18, p = 0.244, $d$ = 0.176, 95% CI mean difference [−0.015, 0.058]), whereas the proportion of touches ending in systole was significantly smaller than chance level ($t$(44) = −3.67, p = 0.0007, $d$ = −0.547, 95% CI mean difference [−0.079, −0.023]).

Last, for the gratings condition, we examined whether the proportion of touches changed as a function of task difficulty. We performed two separate one-way ANOVAs (starting and ending touches) with seven levels (task difficulty). These analyses showed that the proportion of touches starting and ending in each cardiac phase did not vary with task difficulty ($F$(6, 270) = 1.12, p = 0.349, $\eta p^2$ = 0.024) and ($F$(6, 270) = 0.69, p = 0.659, $\eta p^2$ = 0.015).

Overall, these results show that our participants did not move to start the touch in a particular phase of the cardiac cycle. However, they were more likely to end the touch and subsequent sensing in the diastole phase of the cardiac cycle. This is indicated by the smaller proportion of touches ending in systole. This effect was observed both during the condition requiring tactile discrimination and the simple movement control condition. Thus, based on these results we could not conclude that the phase coupling that we observed was related to the perceptual discrimination.

## Proportion of correct responses in each cardiac phase

After examining if our participants tended to generate active touches in a specific cardiac phase, we examined the proportion of correct responses as a function of the phase. To this end, we calculated the proportion of active touches in which participants correctly discriminated the orientation of the gratings. Since in passive sensing studies, greater perceptual sensitivity has been found in diastole (*Al et al., 2020*; *Motyka et al., 2019*), we expected greater performance for trials initiated in this cardiac phase.

Here, we performed an rmANOVA with three variables: cardiac phase (systole, diastole), event type (start, end of touch), and task difficulty (one to seven levels of difficulty). This analysis showed that the effect of cardiac phase ($F$(1, 45) = 1.40, p = 0.243, $\eta p^2$ = 0.03), the effect of event type ($F$(1, 45) = 1.71, p = 0.198, $\eta p^2$ = 0.04), and all interactions between these variables were not significant (p > 0.340). Overall, these results show that regardless of the cardiac phase in which the touches were generated, our participants were equally correct in their responses (*Figure 3B*). Thus, we could not conclude based on these results that phase coupling in active sensing touch confers a perceptual advantage for tactile discrimination. Yet, these results only reflect the proportion of correct responses. A key parameter to-be-analysed is the time that participants spent holding their touches as a function of the cardiac phase in which they initiated them. In the next analysis we tested this.

## Duration of touches initiated in each cardiac phase

Active sensing involves moving to seek information, as well as not moving to extract sense data (fixating the sensor apparatus, i.e., stationary holds). To examine this, we tested whether the duration of stationary holds in active touch sensing depends on the phase of the cardiac cycle in which the

touch is initiated. Specifically, we tested the prediction that if touches were initiated in systole, their duration would be greater to counteract the reduced perceptual tactile sensitivity in systole (*Al et al., 2020*; *Motyka et al., 2019*). Conversely, we expected shorter touches when they were initiated in diastole.

To test this, we first calculated the length of time that participants spent holding their touches when these were initiated in systole or diastole. Then, we compared these durations for the gratings and the control movement condition. Since the latter had no difficulty level (no orientation to-be-discriminated), we compared its duration against that of the gratings data with all levels of difficulty collapsed. This was performed with an rmANOVA, comprising two variables and two levels: experimental condition (gratings, control), and cardiac phase (systole, diastole). The results showed a non-significant effect of cardiac phase ($F$(1, 44) = 3.839, p = 0.056, $\eta p^2$ = 0.001), a significant main effect of condition ($F$(1, 44) = 48.614, p < 0.0001, $\eta p^2$ = 0.509), and a significant interaction between cardiac phase and condition. For the gratings condition, we compared the duration of touches initiated in each phase. A Shapiro–Wilk test indicated a deviation from normality ($w$ = 0.934, p = 0.012). Hence, we used a Wilcoxon signed-rank test to show that the duration of touches initiated in systole 1143 ms (SD = 486.6) was greater than when touches were initiated in diastole 1093 ms (SD = 443.4), $z$ = 855, p = 0.0004, $r$ = 0.582, 95% CI mean difference [18.11, 71.18], *Figure 3C*. For the control movement condition, the equivalent analysis showed no differences in the duration of touches initiated in systole ($M$ = 708, SD = 441.7) and diastole ($M$ = 717, SD = 429.4), $z$ = 461.5, p = 0.531, $r$ = −0.108, 95% CI mean difference [−25.25, 13.50], *Figure 3D*.

Then, for the gratings condition, we examined whether the duration of touches started in each cardiac phase varied with task difficulty. An rmANOVA with the variable cardiac phase (systole, diastole) and task difficulty (one to seven levels of difficulty) was close but did not reach significance ($F$(3.917, 176.267) = 2.43, p = 0.051, $\eta p^2$ = 0.051).

Overall, these results demonstrate that the duration of touches varied as a function of the cardiac phase in which these were initiated. Touches initiated in systole were longer than those initiated in diastole. Given that tactile sensitivity is greater in diastole than in systole (see e.g., *Al et al., 2021a*), these results suggest that our participants overcome the less sensitive phase (systole) through the elongation of the holding touch beyond this phase. Furthermore, and importantly, this modulation was only observed when tactile discrimination was required.

## Variability and duration of touches initiated in each cardiac phase

Above we show that when participants started to touch the gratings in the systole phase, they touched for longer periods (vs. diastole). Importantly, in our task the participants were able to freely start, hold, and end their touches. This is reflected in the range of times that participants spent touching the stimuli. There was considerable variability in how long participants touched the gratings during the task (see *Figures 2B and 3C*). Some participants held for relatively short periods of time and did not modulate their touching duration with task difficulty. In contrast, some participants held for longer periods and increased their touching time with task difficulty. In the current analysis, we explore this between-subject variance. Specifically, we examined if such variability changed as a function of task difficulty and whether it related to the duration of touches initiated in each cardiac phase. Note that this was not a planned analysis as we did not expect before data collection that the duration of participants' touches would vary to the degree it did.

To capture this between-subject variance, we calculated for each participant the standard deviation of how long they touched the gratings (i.e., holding time) across all difficulty levels. Next, we ran the previous analysis again but with this variable as a between-subject covariate. When adjusting the results by this covariate, the main effect of task difficulty in the rmANOVA was still significant ($F$(2.909, 127.993) = 3.070, p = 0.032, $\eta p^2$ = 0.065), and the effect of cardiac phase and its interaction with task difficulty were not significant ($F$(1, 44) = 0.923, p = 0.342, $\eta p^2$ = 0.021) and ($F$(4.089, 179.904) = 0.0896, p = 0.469, $\eta p^2$ = 0.020). In addition, there was a significant interaction between task difficulty and touch variability ($F$(2.909, 127.993) = 41.706, p < 0.0001, $\eta p^2$ = 0.487), a significant interaction between cardiac phase and touch variability ($F$(1, 44) = 17.728, p = 0.0001, $\eta p^2$ = 0.287), and a significant interaction between task difficulty, cardiac phase, and touch variability ($F$(4.089, 179.904) = 4.239, p = 0.002, $\eta p^2$ = 0.088). In addition, the correlation between participants' correct responses and the variability in the duration of their touches was significant (Pearson's $r$ = 0.387, p = 0.008;

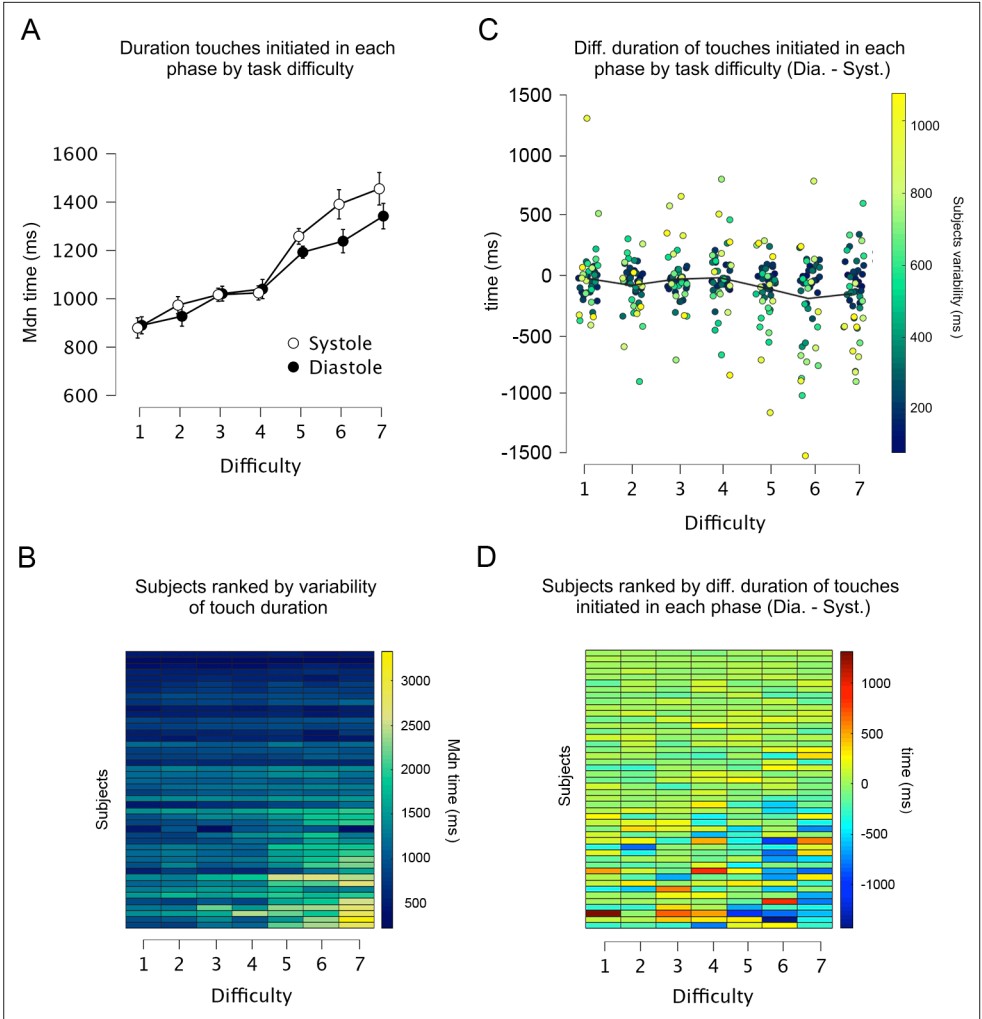

**Figure 4.** Duration and variability of touches initiated in each cardiac phase. (**A**) Across all levels of difficulty (1 easiest, 7 most difficult), the duration of touches when initiated in each cardiac phase. (**B**) Heatmap showing all participants ranked as a function of touch variability in the whole experiment. Each row shows one participant and each column one level of difficulty (e.g., participants at the top varied the least the duration of their touches). The duration of touches changed with task difficulty. (**C**) Differences between the duration of touches started in diastole and systole (Dia. − Syst.) across all levels of difficulty. Each dot represents one participant, and the colouration denotes their variability in the whole experiment (e.g., participants who varied the least are shown as dark blue dots). (**D**) Data from panel C are depicted as a heatmap with participants ranked as a function of touch variability in the whole experiment. Participants who varied the duration of their touches often spent more time touching in systole (vs. diastole) in the higher levels of difficulty (Task difficulty × Cardiac phase × Touch variability: p = 0.002). Error bars depict standard errors; n = 46.

*Supplementary file 1*) indicating that participants who varied the duration of their stationary holds in the experiment were also those with greater tactile discrimination performance.

Overall, these results demonstrate that touches initiated in systole were held for longer periods (vs. touches initiated in diastole). This cardiac-related modulation was particularly exhibited by those participants who varied the duration of their touches when sampling the most difficult gratings (*Figure 4*). This is consistent with the known reduced tactile sensitivity during systole and the idea that our participants held for longer periods to overcome this.

## Cardiac interbeat interval as a function of starting touch

The previous analysis showed that our participants spent more time touching the gratings when these were initially touched in systole, during the cardiac phase with reduced tactile sensitivity. In addition,

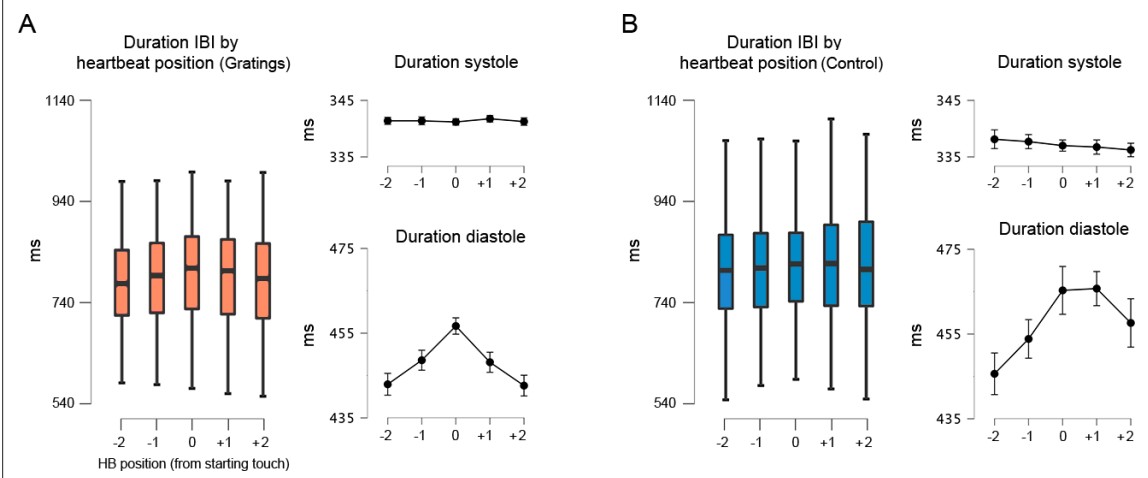

**Figure 5.** Interbeat interval (IBI) before, during, and after the touch of the stimuli. (**A**) For the gratings condition, the IBI of the heartbeat where the touch was initiated (0 on the x-axis) was significantly longer than the preceding and succeeding heartbeats (p < 0.0001). Right panels: with similar statistical effects, this difference was driven by the elongation of the diastole phase of the cardiac cycle; n = 46. (**B**) For control movement condition, the IBI of the heartbeat where the touch was initiated was significantly longer than in the two preceding heartbeats (p < 0.015). Right panels: with similar statistical effects, these differences were driven by the elongation of the diastole phase of the cardiac cycle; n = 45. All p-values adjusted using the Holm–Bonferroni method. The box plots depict the interquartile range (IQR). Given the number of Post Hoc comparissons, see main text and *Supplementary file 1*.

another mechanism that could facilitate tactile perception would be the lengthening of the cardiac cycle as this mostly increases the proportion of the cycle in diastole, which has been associated with the greatest tactile sensitivity. To test this, we first examined changes in the duration of the cardiac cycle for the start of the touch. Second, we examined whether these changes were due to variations in the systole phase or the diastole phase of the cardiac cycle. Accordingly, we calculated the inter-beat intervals (IBIs) of the cardiac cycle in which participants initiated each touch, as well as of the two preceding and subsequent heartbeats (i.e., IBIs relative to heartbeat position).

The rmANOVA comparing the duration of these IBIs between our experimental conditions showed a significant main effect of condition ($F(1, 44) = 14.3$, p = 0.0005, $\eta p^2 = 0.246$), a significant main effect of IBI position ($F(4, 2.164) = 12.5$, p < 0.0001, $\eta p^2 = 0.221$), and a significant interaction between both variables ($F(4, 2.841) = 6.37$, p < 0.0001, $\eta p^2 = 0.126$). Then, we analysed each experimental condition separately. In the gratings condition, there was a significant main effect of heartbeat position on the duration of the IBIs ($F(2.062, 92.788) = 11.65$, p < 0.0001, $\eta p^2 = 0.206$). Post hoc comparisons showed that the duration of the IBI in the starting touch (M = 797.8, SD = 111.7) was significantly longer than in the remaining heartbeats (all p < 0.006, all d > 0.502; M − 2 = 784.2, M − 1 = 789.9, M + 1 = 789.8, M + 2 = 783.7; adjusted for multiple comparisons). For the control movement condition, the results also show a significant main effect of heartbeat position on the duration of the IBIs $F(2.595, 114.201) = 10.75$, p < 0.0001, $\eta p^2 = 0.196$. Post hoc comparisons revealed that the duration of the IBI in the starting touch (M = 821.4, SD = 118.6) was significantly longer than in the two previous heartbeats (all p < 0.015, all d > 0.473; M − 2 = 802.5, M − 1 = 810.6). See *Supplementary file 1C–F* for all post hoc comparisons.

Then we examined whether the above differences were driven by changes in the length of the systole or diastole phase. The results showed that for both stimulus conditions, the duration of systole in the preceding and successive heartbeats (relative to the staring touch) did not change (p > 0.235, $\eta p^2 < 0.032$). Conversely, for the diastole phase, the analyses showed a significant main effect of heartbeat position (p < 0.0001, $\eta p^2 > 0.200$). The post hoc analysis revealed that the changes that we observed in the whole IBIs were due to variations in the diastole phase (*Figure 5A, B*). Moreover, in the gratings condition, these changes covaried with task difficulty; see *Supplementary file 1*.

These results confirm the deacceleration of the cardiac cycle in anticipation or reaction to salient stimuli and the later acceleration after the implementation of a response (*Lacey and Lacey, 1977*; *Sandman et al., 1977*; *Coles and Duncan-Johnson, 1975*; *Park et al., 2014*). Moreover, our results show that changes in the duration of the cardiac cycle are due to the lengthening and compression

of the diastole phase (relative to touch onset). The lengthening of the diastole phase during active sensing touch might aid performance.

## Discussion

The effect of the cardiac phases upon stimulus processing has previously been observed by time-locking the presentation of stimuli to the cardiac phase (e.g., *Al et al., 2021b*; *Critchley and Garfinkel, 2018*; *Garfinkel et al., 2020*). However, in our everyday experiences, sensory information is not encountered in such a phase-locked manner. Here, we tested the following hypothesis: If the phase of the cardiac cycle is an important modulator of perception and cognition, as previously proposed, it should modulate the way in which we freely and actively seek information in the world. To test this, we recorded the ECGs from human participants while they performed a tactile discrimination task of grating orientation. Importantly, they freely initiated, held, and ended the sensing of the stimuli.

From an interoceptive inference framework, studies in passive sensing have shown that tactile sensitivity is reduced during the systole phase of the cardiac cycle (*Al et al., 2021a*; *Grund et al., 2022*; *Motyka et al., 2019*). Hence, to avoid this drop in sensitivity, participants could circumvent the sampling of information in systole. This is precisely what the results of the present work suggest. The duration of subjects' touch (stationary holds) varied as a function of the cardiac phase in which they initiated it. Touches initiated in the systole phase of the cardiac cycle were held for longer periods of time. This is consistent with the idea thar during periods of reduced tactile sensitivity, the participants actively sought to acquire sense data for longer periods of time. Furthermore, we show that this effect was most evident in those participants who varied the length of their touches as a function of task difficulty. In addition, this modulation in the duration of the touches was associated with improved performance. Conversely, while touches in the control condition were coupled to the cardiac cycle, their length was not modulated as a function of when in the cycle these were initiated. This latter condition only required movement, which seems facilitated during systole (*Al et al., 2021a*; *Galvez-Pol et al., 2020a*; *Konttinen et al., 2003*; *Kunzendorf et al., 2019*; *Ohl et al., 2016*; *Palser et al., 2021*). In addition, we showed that after initiating the touch, participants had a significant deceleration of their heart rate. This deceleration was driven by the lengthening of the diastole phase of the cardiac cycle, which has been associated with greater perceptual sensitivity. This cardiac modulation may aid participants' performance.

### Coupling between active sensing and transient bodily cycles

Increasing evidence suggests that ever-fluctuating interoceptive signals exert a putative effect over stimulus processing (*Azevedo et al., 2017*; *Azzalini et al., 2019*; *Azzalini et al., 2021*; *Garfinkel and Critchley, 2016*; *Rae et al., 2018*). This work is often conceptualized within the predictive coding framework (*Allen et al., 2019*; *Friston et al., 2017*) to explain how the brain integrates interoceptive and exteroceptive information to optimize beliefs about the world. Considering this, prevailing models of perception and learning suggest that humans and animals do not only update their beliefs with incoming sensory evidence, but also seek information in a way that maximizes the reduction of uncertainty (*Yang et al., 2016*). This is reflected in the way humans and animals purposely seek information through the movement and control of the sensor apparatus (e.g., whiskers, eyes, fingers, and antennae) in whatever manner best suits the task (*Grant et al., 2014*; *Prescott et al., 2011*; *Yang et al., 2016*).

We found that participants' touch in the control condition was coupled to the cardiac cycle. Yet, its length did not vary as a function of when in the cycle was initiated. This is congruent to recent evidence showing that during simple hand movements, high corticospinal excitability and desynchronization of sensorimotor oscillations are observed in systole (*Al et al., 2021b*). However, we reason that a fine use of our sensorium might entail the coupling of behaviour and quiescent bodily cycles. During the systole phase of the cardiac cycle, pressure sensors located in the carotid sinus, coronary arteries, and aortic arch (baroreceptors), detect changes in blood pressure and convey the strength and timing of heartbeats to the brainstem through the vagus nerve (*Critchley and Harrison, 2013*; *Davos et al., 2002*). Then, the arterial baroreflex system through variations in heart rate, vascular tone, and stroke volume, buffers blood pressure fluctuations (*Vaschillo et al., 2011*; *Vaschillo et al., 2012*). Simultaneously, the ejected blood exerts pulsations with a direct effect on muscle activity

(*Birznieks et al., 2012*; *Fallon et al., 2004*), eliciting small head and eye movements (*Debener et al., 2010*; *Galvez-Pol et al., 2022*; *Lauridsen et al., 2019*). Altogether, these variations could interfere, compete, or block the allocation of attentional and/or representational resources for incoming sensory information. Thus, one possibility is that people actively seek more sensory information during the reduced presence of these signals.

Early evidence for this account, comes from studies showing that elite rifle shooters pull the trigger during diastole, with beginners firing during either phase but with better results during diastole (*Helin et al., 1987*). It has been suggested that this effect is related to the mechanical movement caused by the heart's contraction (*Konttinen et al., 2003*), which is also one principle suggested in the sampling of active visual information during silent cardiac activity (*Galvez-Pol et al., 2020b*). Relatedly, other examples of physiological coupling in humans have been shown in another facet of the cardiovascular system, that is, respiration. *Perl et al., 2019* showed the alignment of self-initiated cognitive tests with the beginning of the inspiratory phase, and recent work by *Kluger et al., 2021* has shown respiration-locked performance in a near-threshold spatial detection task. In this context, *Allen et al., 2022* have proposed a predictive coding interpretation of respiratory cycle effects where breathing rhythmically modulates neural gain, which in turn optimizes cognitive and affective processing.

## Constraints of generality and future work

Here, we consider the constraints of generality proposed by *Simons et al., 2017*. First, the type of tactile discrimination used here (gratings orientation) represents one of the vast spectra of haptic behaviours that people use in daily life. For instance, detection, discrimination, and/or identification of stimulus' texture, size, shape, as well as socially relevant affective touch, which has been linked to the activation of C-tactile fibres and integration of interoceptive information in the insula (*Kirsch et al., 2020*). Second, while we isolated the sense of touch from other exteroceptive modalities such as vision, these and other modalities frequently work together at the sensor and neuronal levels (*Banati et al., 2000*; *Galvez-Pol et al., 2020a*; *Taylor-Clarke et al., 2002*; *Zhou and Fuster, 2000*). Third, our stimuli were judged by young adults (18–35 years old) recruited through a university subject pool. Thus, we expect the results to generalize to situations in which participants engage with similar stimuli. Moreover, we believe the results reported here will be reproducible with young adults from similar subject pools serving as participants. However, we do not have enough evidence to state that our findings will occur outside of these settings, nor to state which changes could be observed when either the stimuli or type of task differ from those reported here. We have no reason to believe that the results depend on other characteristics of the participants and materials.

Future research could examine whether the functional coupling of the sensorium with quiescent periods of the cardiovascular system interacts with different levels of conscious interoceptive inferences. To this aim, further work could examine how active sensing unfolds in individuals who sense, interpret, and integrate to a different extent signals originating from within the body (e.g., interoceptive sensibility; e.g., *Galvez-Pol et al., 2021*; *Garfinkel et al., 2015*; *Palser et al., 2018*; *Suksasilp and Garfinkel, 2022*). Likewise, future work in active sensing may include the co-registration of neuronal and physiological recordings. This might include, for instance, heartbeat, motor-cortical, and/or somatosensory-evoked activity (e.g., *Al et al., 2020*; *Galvez-Pol et al., 2018*; *Galvez-Pol et al., 2020a*; *Montoya et al., 1993*; *Park and Blanke, 2019*; *Candia-Rivera et al., 2022*). As described in *Kluger et al., 2021*, bodily entrained fluctuations in neural activity may represent a mechanism for uniting neural and behavioural findings. While evidence for this account has been accumulating in the animal literature, specially between the respiratory cycle and different behaviours (*Kurnikova et al., 2017*; *Moore et al., 2013*), the field of active sensing in humans with physiological recordings remains largely uncharted. Here, the examination of the heart, an intrinsic oscillator with a faster cycle than respiration, might aid to understand the hierarchical organization of neural oscillations (*Lakatos et al., 2005*) with bodily rhythms such as those originating from the whole cardiovascular system.

Finally, our results indicate that the previously described effects of the cardiac cycle upon perception might apply to temporally constrained tasks, for example, those in which the stimuli are presented for a brief and equal duration during the distinct cardiac phases. While the findings of those tasks have been fundamental to advance the field, we show that in more realistic scenarios, human subjects may overcome cardiac-related effects by actively sensing the stimuli in whatever manner best suits the task (e.g., sensing for longer periods during the cardiac phase with decreased sensitivity). We believe that

our work posits a stimulating framework for further studies at the intersection between exteroception and interoception in more ecological settings.

## Materials and methods

### Participants

Psychophysiological studies and analyses of cardio-sensory processing usually involve sample sizes on the order of 30–50 participants (*Al et al., 2020*; *Galvez-Pol et al., 2020b*; *Grund et al., 2022*; *Herman and Tsakiris, 2021*; *Kunzendorf et al., 2019*; *Motyka et al., 2019*). Accordingly, we recruited 50 healthy adults (32 females; age range = 18–35 years old) who participated in the current study. All participants reported normal cardiac condition, volunteered to take part in the experiment, gave informed consent, and were reimbursed for the participation. Ethical approval for the study was obtained from the UCL research ethics committee. Two participants were excluded due to a faulty recording of their responses. Prior to the further analysis, we tested whether the performance of each participant was significantly greater than chance. To this end, we tested whether participants' performance at each difficulty level was significantly greater/less than chance using a binomial test with a null probability of 0.5 (chance) and a significance level of 0.05. Two participants showed no significant difference from the chance level at any difficulty level, even at the easiest difficulty level. These two participants were excluded from further analysis. In the control condition, one more participant was excluded from the analysis due to a faulty recording of the ECG data. Thus, the data of 46 participants were included in the analysis of the main experimental condition, and 45 in the control condition; these conditions were analysed separately.

### Task and procedure

Participants were seated with the forearm of their dominant hand rested on the top of a table in palm-up position. Participants took part in an active touch sensing paradigm where they were presented with tactile gratings of different widths (narrow to wider gratings; see Stimuli and apparatus). The participants' task was to touch the gratings (one per trial) with the index finger of the dominant hand to determine their orientation. From the participants' perspective, half of these gratings were horizontal and the other half vertical. Specifically, the grids of the gratings were oriented either perpendicular or parallel with the vertical axis of the finger. The gratings were randomly presented through a custom-built device by the experimenter, who was seated opposite to the participants. To determine the orientation, the participants had to move their index finger up, contact the grating, move their finger down, and respond verbally to state the orientation (see Instructions). Then, the experimenter keyed the response. The experimenter seated opposite the participant and was able to see when the participants started touching, for how long they held their touch, and when they finished it. This was visualized as continuous recording, online, and on a screen facing only the experimenter. This screen showed the continuous recording of the touches as pulses in Spike2 8.10 (Cambridge Electronic Design Limited, Cambridge, UK). The amplitude of the pulse departed from zero when a touch was started, and it was maintained for as long as the participants held their touch. Then, it went back to zero amplitude when the touch was finalized. If the participants answered before the pulse went back to zero amplitude, they had responded before releasing the index finger from the tactile stimuli. When this happened, they received feedback, and the entailing trial was discarded. Importantly, once the grating was positioned, the participants were free to initiate, hold, and finalize the touch (i.e., they started, held, and ended touching when they felt like). Therefore, sensation arose through their movement rather than through the passive movement of the gratings.

Before the experiment, the participants received instructions (see Methods) and completed one practice block of 14 trials. If they did not have further questions, they completed 12 more blocks of 14 trials while their ECG and responses were recorded. In the last two blocks, no grating was presented and instead a flat stimulus of the same diameter was used. The participants were instructed to follow the same instructions, but with no need to report any orientation at the end of the trials. This was the movement control stimulus.

## Stimuli and apparatus

We used a custom-built device to present the gratings to the participants. This had two parts: a small platform with a rail, and a sliding section that held fifteen gratings (14 gratings with seven widths by two orientations, and one flat stimulus). The later section could slide along the former and allow presenting one grating at a time through an opening in the centre of the device, above the participants' index finger. The sliding section was moved by the experimenter, who presented the gratings in random order by following a list generated in MATLAB R2016b (The MathWorks Inc, Natick, MA, USA). The device, gratings, and list of gratings to-be-presented were only visible to the experimenter. After each trial, the experimenter keyed the participants' verbal responses in a computer.

The stimuli were made using the Ultimaker-2 3D printer (Ultimaker, Geldermalsen, NL) and were designed using online modelling software (Tinkercad; Autodesk Inc, CA, USA). The gratings consisted of two copies of seven gratings with increasing width of 0.4 mm (from 0.412 to 2.8 mm). Each copy was positioned in the device either vertically or horizontally. When the participants made contact with these stimuli, a continuous pulse at 1000 Hz was sent to a CED Power Unit 1401-3A. This signal was recorded simultaneously with the participants' ECG in Spike2 8.10 (Cambridge Electronic Design Limited, Cambridge, UK). The amplitude of the continuous pulse was set according to the width of the presented grating. Therefore, the pulse itself contained the data required to compute the onset, duration, difficulty, and offset of participants' touches.

## Instructions to the participants

Here, we explicitly indicate the instructions given to the participants: '*In this experiment, we will present to you small gratings with different widths (from narrow to wider gratings). These gratings have two possible orientations: (i) horizontal or (ii) vertical. Here we will randomly present to you one grating at a time (one per trial). Your task is to touch these gratings with your index finger and tell us whether the grating is oriented vertically or horizontally. In each trial, the experimenter will randomly select one of the gratings to be touched. Once the experimenter has placed the grating in position, she/he will place both hands on top of the device (this means that the grating is ready and secured). Then, feel free to touch the grating whenever you feel like, and for as long as you feel like. Yet do not overthink. For this, use your index finger and please produce a single tap ("touching"). Make sure you make firm contact with the grating. Once you touch, do not swing, swipe, or turn your finger (you should "touch-in and touch-out" by just moving your finger up and down). Once you know the orientation, "Touch-out" and only then tell us verbally the orientation of the grating (we will key your response). Please tell us the orientation even if you are uncertain. To get familiar with the task, you will do a practise block. Next, you will do 12 blocks of ~4 mins each. Do you have any questions?*'.

## Data preprocessing and ECG recording

The ECG was recorded using a D360 8-Channel amplifier (Digitimer Ltd, Hertfordshire, UK) in Spike2 8.10; sampling rate 1000 Hz. The ECG electrodes (Skintact Fannin Ltd, Dublin, IE) were placed over the right clavicle and left iliac crest according to Einthoven's triangular arrangement. To investigate whether participants' touch varied along the cardiac cycle, as well as along the cardiac phases comprised in the cycle (systole and diastole), we proceeded to identify the start and end of each phase. The start of each cycle and the systole phase were detected by computing the R-peaks of the QRS complex in the ECG. To this aim, we filtered the ECG (3–30 Hz) using the HEPLAB toolbox implemented in EEGLAB (*Delorme and Makeig, 2004*; *Perakakis, 2019*, respectively). We computed the local maxima of the ECG by using the *findpeaks* function in MATLAB with a minimum peak-to-peak distance of 550 ms. By doing so, we obtained the timepoints of the R-peaks occurring during participants' touches. Next, we identified the end of the systole phase (start of the diastole phase) for each cardiac cycle. The change of phase can be estimated by computing the end of the T-wave in the ECG. This was done by locating the T-peak as a local maximum after the QRS complex and calculating the area of a series of trapeziums along the descending part of the T-wave. The point at which the trapeziums' area is maximal approaches the end of the systole phase in the ECG (*Vázquez-Seisdedos et al., 2011*). We visually inspected the R-peaks and end of the systoles returned by the algorithm and applied small adjustments when required. While it is true that the length of the electrical systole in the ECG cannot be fully equated with the mechanical systole (*Fridericia, 1921*), both are closely

tied under normal conditions (*Fridericia, 1921*; *Boudoulas et al., 1981*; *Gill and Hoffmann, 2010*; *Motyka et al., 2019*).

Trials in which the participants made consecutive contacts (i.e., more than one touch per grating and trial), responded verbally to state the orientation of the grating before finalizing the touch, held their touch less than 100ms, more than 5000 ms, or were ±3 SD of the individuals' mean duration, were excluded from further analysis. Likewise, given the temporal consistency between R-peaks and end of T-waves, trials in which this interval was separated by ± 1.5 × median of the participants' interval in milliseconds were not considered; in total 94.5% of the total number of trials were kept for further analyses.

## Data analysis

### Touch responses as a function of absolute cardiac cycle

To examine whether there was a significant statistical relationship between when participants touched and the cardiac cycle, we analysed the data in two different ways. First, circular statistics were employed to exploit the repeating nature of the cardiac cycle. We calculated the phase of each event of interest (touch initiation, mean contact phase of the touch, and end of touch) as a function of the whole cardiac cycle. For instance, for an R–R interval of time $t_R$, where the start of a touch occurred at time $t_E$, we calculated $t_E/t_R \times 360$. This results in values between 0° and 360° for each event (0 indicating the start of the cardiac cycle). Then, we calculated the participants' mean phase for each event. As in previous research, we tested separately whether the participants' mean phase differed from a uniform distribution using Rayleigh tests (*Al et al., 2020*; *Galvez-Pol et al., 2020a*; *Kunzendorf et al., 2019*; *Motyka et al., 2019*; *Ohl et al., 2016*). We performed this analysis for the start, mean of the holding touch, and end of all touches, separately for the gratings and the control flat stimulus. The mean of holding touch entailed the circular averaging of the whole stationary hold. This often spanned across more than one heartbeat. This analysis was subsequently performed for each level of difficulty.

### Touch responses as a function of cardiac phases

The previous analysis considers the repeating nature of the cardiac cycle, but it does not consider its biphasic nature. Previous studies have shown that responses to stimuli vary as a function of the phase of the cardiac cycle in which information is processed (i.e., systole and diastole; e.g., *Garfinkel et al., 2014*; *Al et al., 2020*; *Leganes-Fonteneau et al., 2020*; *Grund et al., 2022*). In a second analysis, we examined participants' responses as a function of the phase of the cardiac cycle, systole or diastole. For each heartbeat we defined systole as the time between the R-peak and the end of the T-wave. Then, for the same heartbeat, we used the length of this systole window to define a diastole window, which was located at the end of the heartbeat. The length of these windows was used to equate the probability of having an event in the two phases of the cardiac cycle. This approach has advantages compared to a method with fixed time windows (e.g., defining systole as the 300 ms time window following the R-peak) because it accounts for within- and between-subject differences in the length of systole and diastole (i.e., the heart rate); see similar implementation in *Al et al., 2020*; *Grund et al., 2022*; *Motyka et al., 2019*. To determine the end of T-wave, a trapezoidal area algorithm was applied in each trial (see Data preprocessing).

We analysed (1) the proportion of events (start and end touches) occurring during time windows encompassing the systole and diastole phases of the cardiac cycle, and (2) the duration of touches as a function of the cardiac phase in which they were initiated. Note that in the analysis of the proportion of touches by cardiac phase, we only analysed the proportion in one phase (i.e., systole). This was done to reduce the dependency and redundancy in the analysis. For instance, when comparing the proportion of 0.48 touches in systole vs. 0.52 in diastole (adding up to 1) and when comparing these proportions against chance level (0.5) Here, it is only necessary to compute one proportion because it also denotes the proportion and modulation in the other phase. We computed these analyses through rmANOVA and follow-up *T*-tests for all touches, as well as separately for each level of difficulty according to the gratings' width. Mauchly's *W* was computed to check for violations of sphericity, Greenhouse–Geisser adjustments to the degrees of freedom were applied when appropriate, and p values were corrected for multiple comparisons using the Holm–Bonferroni method.

## Data availability and transparency statement

We report $t$ and $p$ values, mean and standard deviations, 95% CIs, and effect sizes, that is, partial eta-squared ($\eta p^2$) and Cohen's $d$. We used in-house and standard code to pre-process and plot our data. Specifically, the pre-processing code was implemented in MATLAB R2018a (The MathWorks Inc, Natick, MA, USA), and the values for analyses and plots were computed in JASP (JASP Team 2021, Version 0.16) and in R using the *circular* package (Version 0.4-93). The code, anonymized data, as well as the resulting analyses are available in the corresponding Open Science Framework repository: https://osf.io/d7x3g/. We report how we determined our sample size, all data exclusions, manipulations, instructions given to the participants, and measures in the study (see Materials and methods). No part of the study procedures was pre-registered before the research was conducted.

# Acknowledgements

We would like to thank all those who participated in and helped to advance this study. This research was supported by the Leverhulme Trust – Grant code RPG-2016-120, United Kingdom (to J.-M.K.) and by the Autonomous Community of the Balearic Islands (CAIB), Postdoctoral grant Margalida Comas Ref PD/036/2019 (to A.G.-P.).

# Additional information

### Funding

| Funder | Grant reference number | Author |
|---|---|---|
| Leverhulme Trust | RPG-2016-120 | James Kilner |
| Autonomus Community of the Balearic Islands, Postdoctoral Grant Margalida Comas | PD/036/2019 | Alejandro Galvez-Pol |

The funders had no role in study design, data collection, and interpretation, or the decision to submit the work for publication.

### Author contributions

Alejandro Galvez-Pol, Conceptualization, Data curation, Software, Formal analysis, Supervision, Funding acquisition, Validation, Investigation, Visualization, Methodology, Writing - original draft, Writing - review and editing; Pavandeep Virdee, Data curation, Visualization; Javier Villacampa, Formal analysis, Visualization; James Kilner, Conceptualization, Resources, Data curation, Software, Formal analysis, Supervision, Funding acquisition, Validation, Investigation, Visualization, Methodology, Project administration, Writing - review and editing

### Author ORCIDs

Alejandro Galvez-Pol http://orcid.org/0000-0002-4430-8336
Pavandeep Virdee http://orcid.org/0000-0002-4732-1472
Javier Villacampa http://orcid.org/0000-0001-9971-9004
James Kilner http://orcid.org/0000-0002-6632-6797

### Ethics

All participants volunteered to take part in the experiment, gave informed consent, and were reimbursed for the participation. Ethical approval for the study was obtained from the University College London research ethics committee ID 10857/002.

### Decision letter and Author response

Decision letter https://doi.org/10.7554/eLife.78126.sa1
Author response https://doi.org/10.7554/eLife.78126.sa2

# Additional files

## Supplementary files
- MDAR checklist
- Supplementary file 1. Post Hoc comparisons and correlational analysis.

## Data availability
All data generated or analysed during this study are included in the manuscript and supporting file. Source data files have been provided for all figures in OSF repository: https://osf.io/d7x3g/.

The following dataset was generated:

| Author(s) | Year | Dataset title | Dataset URL | Database and Identifier |
|---|---|---|---|---|
| Galvez-Pol A, Kilner J | 2022 | Active sensing Touch | https://osf.io/d7x3g/ | Open Science Framework, d7x3g |

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
