## [Editor Report]

This important study investigates the relationship between touch perception and the cardiac cycle. The authors show that people spend more time touching a surface during cardiac systole, where tactile perceptual sensitivity is relatively poor. These findings provide convincing evidence that people actively adjust how they sample sensory information based on internal body states.

---

## [Decision Letter]

**Decision letter after peer review:**

Thank you for submitting your article "Active tactile discrimination is coupled with and modulated by the cardiac cycle" for consideration by *eLife*. Your article has been reviewed by 3 peer reviewers, and the evaluation has been overseen by Andrew Pruszynski as the Reviewing Editor and Floris de Lange as the Senior Editor. The following individuals involved in the review of your submission have agreed to reveal their identity: Nick Holmes (Reviewer #1); Micah Allen (Reviewer #2).

Essential revisions:

As you will see there is a high degree of support and excitement about the study. However, there are extensive detailed comments below that speak to the basic nature of the analytical work and its presentation. It is important that these are carefully considered when preparing the revision. Extensive work is required in this respect.

*Reviewer #1 (Recommendations for the authors):*

Control condition, breathing, speaking

The control condition involved touching an object (for about ~400ms less time than the experimental) "but with no need to report" verbally. This seems to be an important omission. Vocalisation involves breathing, chest movements, and changes in pressure in the chest. As a non-expert, I imagine that the requirement to verbally report something will change the heartbeat. What data do the authors have on this? Why did they choose a non-verbal control? Can the authors be confident their effects are due to the touch and the task rather than just the (verbal) report confound? How does the large difference in touch durations between experimental (~1100ms) and control (~740ms) affect the many time- and phase-dependent conclusions of the paper? the mean cardiac phase also differed a little bit (~790ms vs. 810ms, SD~115ms, so d=.17) – is that relevant? in general, then, the control touch was about one heartbeat long, while the experimental touch was about one-and-a-half. this seems a big difference.

Data removal

Overall, while I am not too concerned about the removal of 5.5% of the data (this is almost universal, sadly – what a waste of global scientific resources!), it is not good practice (see specific comments below). We just need to stop removing so-called 'outliers'. We need to model all our data, and only remove those data that we can be sure reflect genuine technical failure rather than human performance.

Stats

The stats are generally good, but at one point the authors use a measured variable (DV) as if it was a manipulated variable (IV) – the proportion of touches in the systole or diastole phase. This needs fixing; along with my concerns about parametric versus non-parametric analyses (see below).

Results

I feel there are 'too many results'. Just as an example, I did not find the analysis of touch duration variability very well-motivated, and think too much emphasis and analysis has been devoted to it. It feels like a post hoc exploratory analysis that might be better as supplementary information. Along the way there seem to be multiple quite arbitrary decisions being made in order to plot the graphs and describe the data – why theoretically would the SD of touching time be meaningful? why are 4 categories used to colour-code the participants? why are non-parametric analyses used sometimes? I think the authors should take a broad view of the overall results and the 'story' that they are telling and consider whether these (seemingly post hoc) analyses are really contributing. I admit to being confused by them. One clear example where the wealth of results does not seem to help to understand is in the heartbeat durations which are longer for the closest one to touch initiation. The effect looks strong statistically, but it looks the same in the control condition and there is no direct comparison between experimental and control.

One clear strategy to simplify and improve this paper would be only to report statistics where the experimental data are compared directly and in meaningful ways with the control data, rather than reporting independent, parallel analyses of experimental and control conditions. Why run a control condition at all if it is not used to test the experimental condition against? I suspect that some critical result, somewhere, came out the 'wrong' way in this comparison, and it is being 'hidden' somehow – a skeptical suspicion, not an accusation!

*Reviewer #2 (Recommendations for the authors):*

I think this is an excellent article, and I have no major outstanding suggestions. It could be nice to see a supplementary or control analysis regarding the robustness of some of the effects of non-normality, and I am curious to see if anything is going on with the choice accuracy, but I believe such secondary analyses are beyond the scope of an *eLife* revision. Overall this is an excellent manuscript!

*Reviewer #3 (Recommendations for the authors):*

Below I present my specific comments/questions:

• Setup: The authors use custom-built equipment to record and match the timings of tactile events to the ECG signal. Could the Authors provide any data on the accuracy and reliability of this setup (e.g. delays etc)?

• Circular analysis: The Authors used circular analysis to compare the onsets and offsets of touch in relation to the cardiac cycle. However, recently such an approach to the cardiac cycle has been questioned as it "stretches" all points in the cardiac cycle equally, which may not necessarily be physiologically valid (Sherman et al., 2022). In fact, in a re-analysis of the Authors' published dataset with an updated set of statistical techniques, Sherman et al., were not able to replicate the oculomotor events occurring more often in certain cardiac phases. Thus, it raises the issue of whether using an updated circular analysis would yield the same findings.

• It is clear to me how the onsets and offsets of tactile events were analysed using the circular analysis, but the holding time is not clear to me. Was just the middle point at which participants touched the grating taken into account?

• Biphasic analysis: The Authors computed the proportion of events (start and end of touches) occurring during the 2 systolic and diastolic windows with the diastolic window equal in length to the systolic one and positioned at the end of the cardiac cycle. I was wondering what is the reason for this arbitrary widow as opposed to using the whole diastole. By arbitrarily "clipping" the diastole, some trials, those that occurred/ended in early diastole, will not be taken into account in the analysis. One could utilize the whole duration of diastole, the proportions simply need to be then normalized to the proportion of the subject-specific phase length in the total cardiac cycle (see e.g. Kunzendorf et al., 2019 for details).

---

## [Author Response]

Essential revisions:As you will see there is a high degree of support and excitement about the study. However, there are extensive detailed comments below that speak to the basic nature of the analytical work and its presentation. It is important that these are carefully considered when preparing the revision. Extensive work is required in this respect.Reviewer #1 (Recommendations for the authors):Control condition, breathing, speakingThe control condition involved touching an object (for about ~400ms less time than the experimental) "but with no need to report" verbally. This seems to be an important omission. Vocalisation involves breathing, chest movements, and changes in pressure in the chest. As a non-expert, I imagine that the requirement to verbally report something will change the heartbeat. What data do the authors have on this? Why did they choose a non-verbal control? Can the authors be confident their effects are due to the touch and the task rather than just the (verbal) report confound? How does the large difference in touch durations between experimental (~1100ms) and control (~740ms) affect the many time- and phase-dependent conclusions of the paper? the mean cardiac phase also differed a little bit (~790ms vs. 810ms, SD~115ms, so d=.17) – is that relevant? in general, then, the control touch was about one heartbeat long, while the experimental touch was about one-and-a-half. this seems a big difference.

Thank you for this observation. This is an interesting question. As stated in the instructions for the participants (see Instructions), we asked our participants to follow these directions: ‘*Once you know the orientation, ‘Touch-out’ and only then tell us verbally the orientation of the grating (we will key your response)’.* The participants were able to practise these instructions in 14 practice trials before the experiment. During the experiment, the experimenter who seated opposite the participant, was able to see when the participants started touching, for how long they held their touch, and when they finished touching the gratings, with verbalizations happening after the touch. As the reviewer notes any possible confound with the vocalisation can only explain any differences between the ‘control’ and ‘touch’ conditions. It cannot explain any of the effects observed within the ‘touch’ condition as a vocalisation was made on each of these trials which were the main focus of this study.

In the review process what has become clear was that we did not explain clearly the rational for the ‘control’ condition. The main aim of the work described in this paper was to test the hypothesis that in a free active touch condition the timing and/or the duration of the touch would vary as a function of the phase of the cardiac cycle in which the touch was initiated. This hypothesis was based on previous studies that shown modulation in tactile perception as a function of cardiac cycle, with perception being worse in systole than diastole (Al et al., 2020, 2021). However, we were concerned that any effects we might have observed could be confounded by the fact that an active touch requires a movement. For this reason, we additionally ran a movement control condition without tactile discrimination. The control condition was by design devoid of a tactile discrimination element. To make this clearer in the revised version, this is now labelled the ‘movement control condition’.

Data removalOverall, while I am not too concerned about the removal of 5.5% of the data (this is almost universal, sadly – what a waste of global scientific resources!), it is not good practice (see specific comments below). We just need to stop removing so-called 'outliers'. We need to model all our data, and only remove those data that we can be sure reflect genuine technical failure rather than human performance.

Thank you, we appreciate the reviewer’s insight.

StatsThe stats are generally good, but at one point the authors use a measured variable (DV) as if it was a manipulated variable (IV) – the proportion of touches in the systole or diastole phase. This needs fixing; along with my concerns about parametric versus non-parametric analyses (see below).

Thank you for this observation. Indeed, in the previous version of the manuscript we analysed the proportion of touches that started and ended in each cardiac phase. For instance, 0.48 in systole and 0.52 in diastole (adding up to 1) in a rmANOVA with factor cardiac phase and two levels (systole, diastole). We totally agree with the reviewer: this was wrong. In the revised version we have corrected this.

As the reviewer suggests further in the review, analysing the proportion of touches only in one phase is the correct approach (e.g., proportion in systole). In the new results version of the manuscript (‘Start and end of touches as a function of cardiac phases’) we have (i) analysed the proportion of touches in one phase, (ii) explained the rationale behind this, (iii) simplified the plots (the raincloud plots do not show the median, but the mean), and (iv) compared the discrimination task and the control task (as suggested by the reviewer later in the response letter). Please see the new text below; located at the start of the results in the ms.

*Line 221:* “Proportion of touches starting and ending in each cardiac phase

To study the coupling between the movement of the sensor to acquire sense data and the cardiac phase, we examined whether our participants started and ended their active touches in a specific cardiac phase. Based on previous research, we hypothesized a greater proportion of touches starting in systole, and a greater proportion of touches ending at the end of diastole; the latter being the cardiac phase with greater perceptual sensitivity (Galvez-Pol et al., 2020; Kunzendorf et al., 2019; Ohl et al., 2016, Al et al., 2020; Motyka et al., 2019). Importantly, since the proportion of touches in each cardiac phase reflects and depends on the proportion in the other phase (See Data Analysis), we only analysed the proportion of touches in one phase (in systole). Here two different set of analyses were performed.

First, we compared the proportion of touches that started and ended in each cardiac phase between our experimental conditions. Since the control movement condition had no difficulty level (no orientation to-be-discriminated), we compared it against the gratings data collapsed across all levels of difficulty. Here we performed a rmANOVA with two variables and two levels: experimental condition (gratings, control) and event type (starting and ending touches in systole). This analysis showed a significant main effect of event type (*F*(1, 44) = 9.91, *p* = 0.003, *ηp2* = 0.184), a non-significant effect of experimental condition (*F*(1, 44) = 0.47, *p* = 0.499, *ηp2* = 0.010), and a non-significant interaction between both variables (*F*(1, 44) = 4.04, *p* = 0.051, *ηp2* = 0.084). However, we found a stronger trend for the control movement condition (See Figure 3A). For the gratings, the proportion of touches that started in systole was significantly smaller than the proportion of touches that ended in systole (*t*(45) = -2.22, *p* = 0.032, *d* = -0.327, 95% CI mean difference [-0.040, -0.002]). Likewise, in the control movement condition, the proportion of touches that started in systole was significantly smaller than that of touches that ended in systole (*t*(44) = -2.75, *p* = 0.009, *d* = -0.410, 95% CI mean difference [-0.125, -0.019]).

The previous analysis does not test whether the proportion of touches in each cardiac phase is different to what would be expected by chance. We examined this by comparing the proportions of touches that started and ended in each phase against chance (0.5) using onesample T-tests. For the gratings condition, the proportion of touches starting in systole did not differ from chance level (*F*(1, 45) = 0.19, *p* = 0.662, *ηp2* = 0.004). In contrast, the proportion of touches ending in systole was significantly smaller than chance level (*F*(1, 45) = 6.33, *p* = 0.015, *ηp2* = 0.123). Similarly, in the control movement condition the proportion of touches starting in systole did not differ from chance level (*t*(44) = 1.18, *p* = 0.244, *d* = 0.176, 95% CI mean difference [-0.015, 0.058]), whereas the proportion of touches ending in systole was significantly smaller than chance level (*t*(44) = -3.67, *p* = 0.0007, *d* = -0.547, 95% CI mean difference [-0.079, -0.023]).

Last, for the gratings condition, we examined whether the proportion of touches changed as a function of task difficulty. We performed two separate one-way ANOVAs (starting and ending touches) with 7 levels (task difficulty). These analyses showed that the proportion of touches starting and ending in each cardiac phase did not vary with task difficulty (*F*(6, 270) = 1.12, *p* = 0.349, *ηp2* = 0.024) and (*F*(6, 270) = 0.69, *p* = 0.659, *ηp2* = 0.015).

Overall, these results show that our participants did not move to start the touch in a particular phase of the cardiac cycle. However, they were more likely to end the touch and subsequent sensing in the diastole phase of the cardiac cycle. This is indicated by the smaller proportion of touches ending in systole. This effect was observed both during the condition requiring tactile discrimination and the simple movement control condition. Thus, based on these results we could not conclude that the phase coupling that we observed was related to the perceptual discrimination.”

ResultsI feel there are 'too many results'. Just as an example, I did not find the analysis of touch duration variability very well-motivated, and think too much emphasis and analysis has been devoted to it. It feels like a post hoc exploratory analysis that might be better as supplementary information. Along the way there seem to be multiple quite arbitrary decisions being made in order to plot the graphs and describe the data – why theoretically would the SD of touching time be meaningful? why are 4 categories used to colour-code the participants? why are non-parametric analyses used sometimes? I think the authors should take a broad view of the overall results and the 'story' that they are telling and consider whether these (seemingly post hoc) analyses are really contributing. I admit to being confused by them. One clear example where the wealth of results does not seem to help to understand is in the heartbeat durations which are longer for the closest one to touch initiation. The effect looks strong statistically, but it looks the same in the control condition and there is no direct comparison between experimental and control.One clear strategy to simplify and improve this paper would be only to report statistics where the experimental data are compared directly and in meaningful ways with the control data, rather than reporting independent, parallel analyses of experimental and control conditions. Why run a control condition at all if it is not used to test the experimental condition against? I suspect that some critical result, somewhere, came out the 'wrong' way in this comparison, and it is being 'hidden' somehow – a skeptical suspicion, not an accusation!

We thank the reviewer for these observations. Indeed, after this iteration we have noticed that some of the sections were not clear enough (i.e., rationale behind the analysis, all of which were based on previous studies in this field (with the exception of the touch duration variability which was exploratory as was stated in the original version)). Overall, and as the reviewer noted later, it is important that each section can stand by itself. In this sense we have improved each result section by stating the necessity of each analysis and the predictions we had. Also, we have simplified some of the analysis and figures (as shown in the previous reply). We will respond to the specific points as they are detailed below:

About the number of analyses: we believe that the rationale for some of the analysis was not strong enough. In the new version of the manuscript, we have improved the rationale and clarity behind the analyses. In addition, we simplified some of the analyses and corresponding figures (see above reply about using the proportion of touches in one cardiac phase, avoiding redundancy). Please, see new text at the end of this reply.

About the analysis of touch variability: a crucial element of this manuscript is that we asked participants to sample the stimuli at their own pace: purposively seeking information through the movement and control of the sensor apparatus in whatever manner best suits the task. In such an approach (*vs.* experiments that present stimuli for a fixed amount of time), we observed a lot of variability in time participants spent sensing the stimuli as a function of task difficulty. This was not something that we predicted at the start of the experiment. However, in an exploratory analysis we decided to test whether participants’ differences in their sensing-approach (i.e., time) as a function of task difficulty had a significant impact on their cardiac related responses To show this, we have been transparent: (i) all data, graphs, and code are from the start in OSF; (ii) we show the data both at the descriptive (subjects colour coded and ranked by variability) and inferential levels.

About the use of 4 categories to colour-code the participants: we agree with the reviewer. This was our mistake, and it has been corrected. In the new version of the manuscript, the scatter plots of the figure (Figure 4C) are now coloured using a continuous scale.

About pre-planned and post hoc analysis: to be clear – all the analyses were preplanned with one exception, the variability of the duration of the touch as a covariate. As active sensing and physiology in humans is quite novel, we are constantly learning and understanding the data in more meaningful ways —we believe this is a natural phenomenon that comes with advances in new research programmes. We have made this clearer in the revised version of the manuscript, stating with analyses were pre-planned and more exploratory. Also, we more clearly describe which previous studies our analyses are based on. As the reviewer will note that all the data, analysis, and code were and are freely available on the OSF website; we have nothing to hide J. In addition, where appropriate we have now included additional analyses as requested comparing the touch discrimination condition with the movement control condition. This is only possible when we first averaged across difficulty levels as there is only one ‘level’ for the movement control task, i.e., collapsing all levels of difficulty in the discrimination task and comparing this to the control task where no discrimination was required; see new text at the end of this reply.

Line 80 – improving rationale: “Here we tested whether this cardiac-related modulation affects the timing and duration of active touch. Specifically, we tested two hypotheses. First, whether the time of onset and/or offset of the touches reflects sampling of the stimulus during periods with greater tactile sensitivity. In other words, do subjects preferentially touch during the diastole phase of the cardiac cycle? Second, we tested whether the duration of active touch is modulated by the phase of the cardiac cycle in which the touch is initiated. The prediction was that touches initiated in systole would be longer than those initiated in diastole because of the reduced tactile sensitivity in systole.”

Line 164 – improving rationale: “Studies in active sensing have shown the start of movements to acquire sense data occurs during the early period of the cardiac cycle, whereas the actual sensing has been observed during the mid and later periods (Kunzendorf et al., 2019; GalvezPol et al., 2020). In addition, studies in passive sensing have shown greater perceptual sensitivity during these latter periods of the cycle (Al, et al., 2021; Grund et al., 2022; Motyka et al., 2019). Hence, we predicted that the start of active touches would occur in the early period of the cardiac cycle, and the sensing touch (i.e., stationary hold) would preferentially occur during the later and more sensitive period of the cardiac cycle.”

Line 221 – improving rationale: “To study the coupling between the movement of the sensor to acquire sense data and the cardiac phase, we examined whether our participants started and ended their active touches in a specific cardiac phase. Based on previous research, we hypothesized a greater proportion of touches starting in systole, and a greater proportion of touches ending at the end of diastole; the latter being the cardiac phase with greater perceptual sensitivity (Galvez-Pol et al., 2020; Kunzendorf et al., 2019; Ohl et al., 2016, Al et al., 2020; Motyka et al., 2019). Importantly, since the proportion of touches in each cardiac phase reflects and depends on the proportion in the other phase (See Data Analysis), we only analysed the proportion of touches in one phase (in systole). Here two different set of analyses were performed.”

Line 270 – improving rationale: “After examining if our participants tended to generate active touches in a specific cardiac phase, we examined the proportion of correct responses as a function of the phase. To this end, we calculated the proportion of active touches in which participants correctly discriminated the orientation of the gratings. Since in passive sensing studies, greater perceptual sensitivity has been found in diastole (Al et al., 2020; Motyka et al., 2019), we expected greater performance for trials initiated in this cardiac phase.”

Line 290 – improving rationale: “Active sensing involves moving to seek information, as well as not moving to extract sense data (fixating the sensor apparatus, i.e., stationary holds). To examine this, we tested whether the duration of stationary holds in active touch sensing depends on the phase of the cardiac cycle in which the touch is initiated. Specifically, we tested the prediction that if touches were initiated in systole, their duration would be greater to counteract the reduced perceptual tactile sensitivity in systole (Al et al., 2020; Motyka et al., 2019). Conversely, we expected shorter touches when they were initiated in diastole.”

Line 339 – improving rationale of our variability analysis: “Above we show that when participants started to touch the gratings in the systole phase, they touched for longer periods (*vs*. diastole). Importantly, in our task the participants were able to freely start, hold, and end their touches. This is reflected in the range of times that participants spent touching the stimuli […]

Specifically, we examined if such variability changed as a function of task difficulty and whether it related to the duration of touches initiated in each cardiac phase. Note that this was not a planned analysis as we did not expect before data collection that the duration of participants’ touches would vary to the degree it did.”

Line 346 – improving rationale and indicating which analysis was exploratory: “Specifically, we examined if such variability changed as a function of task difficulty and whether it related to the duration of touches initiated in each cardiac phase. Note that this was not a planned analysis as we did not expect before data collection that the duration of participants’ touches would vary to the degree it did.”

Line 389 – improving rationale of analysis of heartbeats duration: “In addition, another mechanism that could facilitate tactile perception would be the lengthening of the cardiac cycle as this would increase the proportion of the cycle in diastole, which has been associated with the greatest tactile sensitivity. To test this, we first examined changes in the duration of the cardiac cycle for the start of the touch. Second, we examined whether these changes were due to variations in the systole phase or diastole phase of the cardiac cycle.”

Line 231 – comparing our experimental conditions: “First, we compared the proportion of touches that started and ended in each cardiac phase between our experimental conditions. Since the control movement condition had no difficulty level (no orientation to-bediscriminated), we compared it against the gratings data collapsed across all levels of difficulty. Here we performed a rmANOVA with two variables and two levels: experimental condition (gratings, control) and event type (starting and ending touches in systole). This analysis showed a significant main effect of event type (*F*(1, 44) = 9.91, *p* = 0.003, *ηp2* = 0.184), a non-significant effect of experimental condition (*F*(1, 44) = 0.47, *p* = 0.499, *ηp2* = 0.010), and a non-significant interaction between both variables (*F*(1, 44) = 4.04, *p* = 0.051, *ηp2* = 0.084).

However, we found a stronger trend for the control movement condition (See Figure 3A).”

Line 298– comparing our experimental conditions: “Then, we compared these durations for the gratings and the control movement condition. Since the latter had no difficulty level (no orientation to-be-discriminated), we compared its duration against that of the gratings data with all levels of difficulty collapsed. This was performed with a rmANOVA, comprising two variables and two levels: experimental condition (gratings, control), and cardiac phase (systole, diastole). The results showed a non-significant effect of cardiac phase (*F*(1, 44) = 3.839, *p* = 0.056 , *ηp^2^* = 0.001), a significant main effect of condition (*F*(1, 44) = 48.614, *p* < 0.0001 , *ηp^2^* = 0.509), and a significant interaction between cardiac phase and condition. For the gratings condition, we compared the duration of touches initiated in each phase. A Shapiro-Wilk test indicated a deviation from normality (*w* = 0.934, *p* = 0.012). Hence, we used a Wilcoxon signed-rank test to show that the duration of touches initiated in systole 1143ms (SD = 486.6) was greater than when touches were initiated in diastole 1093ms (SD = 443.4), *Z* = 855, *p* = 0.0004, *r* = 0.582, 95% CI mean difference [18.11, 71.18], Figure 3C. For the control movement condition, the equivalent analysis showed no differences in the duration of touches initiated in systole (M = 708, SD = 441.7) and diastole (M = 717, SD = 429.4), *Z* = 461.5, *p* = 0.531, *r* = -0.108, 95% CI mean difference [-25.25, 13.50], Figure 3D.”

Line 397 – comparing our experimental conditions: “The rmANOVA comparing the duration of these IBIs between our experimental conditions showed a significant main effect of condition (*F*(1, 44) = 14.3, *p* = 0.0005, *ηp2* = 0.246), a significant main effect of IBI position (*F*(4, 2.164) = 12.5, *p* < 0.0001, *ηp2* = 0.221), and a significant interaction between both variables (*F*(4, 2.841) = 6.37, *p* < 0.0001, *ηp2* = 0.126)."

Reviewer #2 (Recommendations for the authors):I think this is an excellent article, and I have no major outstanding suggestions. It could be nice to see a supplementary or control analysis regarding the robustness of some of the effects of non-normality, and I am curious to see if anything is going on with the choice accuracy, but I believe such secondary analyses are beyond the scope of an eLife revision. Overall this is an excellent manuscript!

Thank you, we appreciate these suggestions. The truth is that the current experiment has lots of potential for reanalysis in various forms. We believe that we have implemented the most efficient and relevant, but more steps could be implemented. In such a case, the whole dataset, code, and JASP files are uploaded in the OSF repository. We hope this help other researchers to explore the data in new and stimulating ways.

Reviewer #3 (Recommendations for the authors):Below I present my specific comments/questions:• Setup: The authors use custom-built equipment to record and match the timings of tactile events to the ECG signal. Could the Authors provide any data on the accuracy and reliability of this setup (e.g. delays etc)?

Thank for this observation. We think that we did not express well-enough what we meant here, and we do apologise. The setup for the recording of the ECG and behavioural responses was standard and based on equipment that we often used in the lab (also used for studies using EEG, TMS, etc). When the participants made contact with the stimuli, a continuous pulse at 1000Hz was sent to a CED Power Unit 1401-3A. This signal was recorded simultaneously with the participants’ ECG in Spike2 8.10 (Cambridge Electronic Design Limited, Cambridge, UK). The only parts of the equipment that were custom-built were the gratings, flat stimulus, and the rail/structure to support them. These were built using a 3D printer. The wording of one of the sections has been modified. We explain better the whole setup. Please see new below.

Line 582: "The experimenter seated opposite the participant and was able to see when the participants started touching, for how long they held their touch, and when they finished it. This was visualized as continuous recording, online, and on a screen facing only the experimenter. This screen showed the continuous recording of the touches as pulses in Spike2 8.10 (Cambridge Electronic Design Limited, Cambridge, UK). The amplitude of the pulse departed from zero when a touch was started, and it was maintained for as long as the participants held their touch. Then, it went back to zero amplitude when the touch was finalised. If the participants answered before the pulse went back to zero amplitude, they had responded before releasing the index finger from the tactile stimuli. When this happened, they received feedback, and the entailing trial was discarded."

Line 613 – not new text, just shown here to indicate presence of these details: The stimuli were made using the Ultimaker-2 3D printer (Ultimaker, Geldermalsen, NL) and were designed using online modelling software (Tinkercad; Autodesk Inc, California, US).

Line 617 – not new text, just shown here to indicate presence of these details: When the participants made contact with these stimuli, a continuous pulse at 1000Hz was sent to a CED Power Unit 1401-3A. This signal was recorded simultaneously with the participants’ ECG in Spike2 8.10 (Cambridge Electronic Design Limited, Cambridge, UK).

• Circular analysis: The Authors used circular analysis to compare the onsets and offsets of touch in relation to the cardiac cycle. However, recently such an approach to the cardiac cycle has been questioned as it "stretches" all points in the cardiac cycle equally, which may not necessarily be physiologically valid (Sherman et al., 2022). In fact, in a re-analysis of the Authors' published dataset with an updated set of statistical techniques, Sherman et al., were not able to replicate the oculomotor events occurring more often in certain cardiac phases. Thus, it raises the issue of whether using an updated circular analysis would yield the same findings.

Thank you for these observations. The circular averaging does not consider the biphasic nature of the cardiac cycle (systole, diastole). This is the main scope of our manuscript. We started by implementing the circular approach to be consistent with the most imminent past literature. Yet, we quickly note this constrain and move forward to analyses and comparisons between conditions using the cardiac phases. This point has been strengthened in the last version of the manuscript.

Line 197: *“*These results indicate that active touches were coupled with the cardiac cycle in the control movement condition. Similar findings have been found in active visual tasks that did not require perceptual discrimination (see Kunzendorf et al., 2019; Galvez-Pol et al., 2020). In a separate analysis of the gratings condition, we did not find evidence of coupling when our participants sought to sense and discriminate the gratings. Yet, while the present analysis considers the iterative nature of the cardiac cycle, it does not reflect its true biphasic physiological nature (the presence of two cardiac phases: systole and diastole) nor differences in how participants performed the task. We examined these in the following analyses.”

Regarding the paper of Sherman et al., 2022, there was an analytical error in the implementation of their analysis. Specifically, in the implementation of the permutation that allows to link events (saccades, touches, etc) and time bins. This has been recognised by the authors and a correction will shortly appear in press. If it is of interest to the reviewer, we are happy to share a piece of code that demonstrate this. Note, also that Sherman et al., do not actually fail to replicate the result. In fact, they perform a different analysis of the data that divides the cardiac cycle into multiple bins and tests for a modulation of bins. I appreciate this might seem like a subtle point but it is important to stress that Sherman did not show a failed replication, the analysis was different and potentially less statistically sensitive.

• It is clear to me how the onsets and offsets of tactile events were analysed using the circular analysis, but the holding time is not clear to me. Was just the middle point at which participants touched the grating taken into account?

Thank you for this observation. We realised that we were not clear enough in the original submission. These are indeed circular means. For the start and end of touches, the circular means entail the first and last concomitant heartbeat, i.e., relative to the start and end of the touch. For the mean of holding touches, it entails the circular averaging of the whole stationary hold. This often spanned across more than one heartbeat. When this is calculated and depicted in circular space might give the impression of entailing a single heartbeat. We have now emphasized/noted this information in the manuscript. Please see the new text below:

Line 179: “The mean phase of the holding touch occurred on average circa the mid part of the cycle at 202°. Yet, it should be noted that holding touches often spanned across more than one cycle.”

Line 213: “Circular means showing the distribution of the start, holding, and end of touches for the grating stimuli.”

Line 685: “The mean of holding touch entailed the circular averaging of the whole stationary hold. This often spanned across more than one heartbeat.”

• Biphasic analysis: The Authors computed the proportion of events (start and end of touches) occurring during the 2 systolic and diastolic windows with the diastolic window equal in length to the systolic one and positioned at the end of the cardiac cycle. I was wondering what is the reason for this arbitrary widow as opposed to using the whole diastole. By arbitrarily "clipping" the diastole, some trials, those that occurred/ended in early diastole, will not be taken into account in the analysis. One could utilize the whole duration of diastole, the proportions simply need to be then normalized to the proportion of the subject-specific phase length in the total cardiac cycle (see e.g. Kunzendorf et al., 2019 for details).

Thank you, we appreciate the reviewer’s remark. Most studies that examine the effect of the cardiac cycle upon perception have used time windows of a fixed length. For instance, they define systole as a time window of ~300ms from the R-peak, and the diastole as a time window of the same length 500ms after the R-peak. However, this approach does not consider that the length of these cardiac phases changes as a function of heart rate. Therefore, this approach might fit some participants while not properly fit others. For this reason, the most direct and related studies (in tactile sensitivity and interoception) have used the method that we use in the current manuscript, i.e., locating the systole and diastole phases, and creating time windows of similar length; see e.g., Grund et al., 2002 (JoN), or Motyka et al., 2019 (Psychophysiol), and Al et al., (2020) PNAS. Indeed, our analysis and time windows was based on these prior studies.

Therefore, the method we used is based on those used in most contemporary and related papers to our research. In this context, this allows (i) examining phasic changes as function of changes in heart rate, and (ii) equates the data to-be-analysed with no further transformations. Also, please note that the work of Kunzendorf et al., (2019) leaves certain time between the systole and diastole phase of the cardiac cycle, which is effectively like the current method. In this context, it is also true that we did not explain the main reason for the choice of method, and that we should have referenced more appropriately this in the right locations along the manuscript. Following this, we have made these changes (clearer text and references) in the updated version of the manuscript. Please see text below.

Line 112: “We analysed the proportion of events (start and end of touches) in two time windows that encompassed the systole and diastole phases of the cardiac cycle (see e.g., Al et al., 2020; Grund et al., 2022; Motyka et al., 2019).”

Line 127: “We used the systole length of each cardiac cycle to define a time window of equal length (to equate events’ probability in diastole; see e.g., Al et al., 2020).”

Line 696: “For each heartbeat we defined systole as the time between the R-peak and the end of the T-wave. Then, for the same heartbeat, we used the length of this systole window to define a diastole window, which was located at the end of the heartbeat. The length of these windows was used to equate the probability of having an event in the two phases of the cardiac cycle. This approach has advantages compared to a method with fixed time windows (e.g., defining systole as the 300-ms time window following the R-peak) because it accounts for within- and between-subject differences in the length of systole and diastole (i.e., the heart rate); see similar implementation in Al et al., (2020); Grund et al., (2022); Motyka et al., (2019).”